# Two Motors and One Spring: Hypothetic Roles of Non-Muscle Myosin II and Submembrane Actin-Based Cytoskeleton in Cell Volume Sensing

**DOI:** 10.3390/ijms22157967

**Published:** 2021-07-26

**Authors:** Nadezhda Barvitenko, Muhammad Aslam, Alfons Lawen, Carlota Saldanha, Elisaveta Skverchinskaya, Giuseppe Uras, Alessia Manca, Antonella Pantaleo

**Affiliations:** 1Independent Researcher, 191014 Saint-Petersburg, Russia; 2Department of Internal Medicine I, Experimental Cardiology, Justus Liebig University, 35392 Giessen, Germany; muhammad.aslam@physiomed.jlug.de; 3Department of Biochemistry and Molecular Biology, School of Biomedical Sciences, Monash University, Clayton, VIC 3800, Australia; alfons.lawen@monash.edu; 4Institute of Biochemistry, Institute of Molecular Medicine, Faculty of Medicine University of Lisbon, 1649-028 Lisboa, Portugal; carlotasaldanha@fm.ul.pt; 5Sechenov Institute of Evolutionary Physiology and Biochemistry, 194223 St. Petersburg, Russia; lisarafail@mail.ru; 6Department of Clinical and Movement Neurosciences, Institute of Neurology, University College London, London NW3 2PF, UK; g.uras@ucl.ac.uk; 7Department of Biomedical Science, University of Sassari, Viale San Pietro 43/B, 07100 Sassari, Italy; alessia_manca@hotmail.it

**Keywords:** cell volume, swelling, shrinkage, non-muscle myosin II, actin cortex, actin polymerization, mechanosensors, proliferation, apoptosis, migration

## Abstract

Changes in plasma membrane curvature and intracellular ionic strength are two key features of cell volume perturbations. In this hypothesis we present a model of the responsible molecular apparatus which is assembled of two molecular motors [non-muscle myosin II (NMMII) and protrusive actin polymerization], a spring [a complex between the plasma membrane (PM) and the submembrane actin-based cytoskeleton (smACSK) which behaves like a viscoelastic solid] and the associated signaling proteins. We hypothesize that this apparatus senses changes in both the plasma membrane curvature and the ionic strength and in turn activates signaling pathways responsible for regulatory volume increase (RVI) and regulatory volume decrease (RVD). During cell volume changes hydrostatic pressure (HP) changes drive alterations in the cell membrane curvature. HP difference has opposite directions in swelling versus shrinkage, thus allowing distinction between them. By analogy with actomyosin contractility that appears to sense stiffness of the extracellular matrix we propose that NMMII and actin polymerization can actively probe the transmembrane gradient in HP. Furthermore, NMMII and protein-protein interactions in the actin cortex are sensitive to ionic strength. Emerging data on direct binding to and regulating activities of transmembrane mechanosensors by NMMII and actin cortex provide routes for signal transduction from transmembrane mechanosensors to cell volume regulatory mechanisms.

## 1. Introduction

Cell fate and functioning, e.g., proliferation, migration, apoptosis, are accompanied and critically determined by appropriate changes in cell volume [1,2,3,4,5,6,7,8]. Changes occurring in the cell under exposure to osmotic challenge consist of modifications in the membrane curvature and alterations in intracellular ionic strength. In response to shrinkage the cell induces regulatory volume increase (RVI) which involves activation of Na^+^-K^+^-2Cl^−^ cotransporters (NKCC), Na^+^/H^+^ exchangers (NHE), Na^+^ channels and uptake of organic osmolytes. Cell swelling leads to regulatory volume decrease (RVD) mediated by activation of K^+^ and Cl^−^ ion channels, K^+^-Cl^−^ cotransporters (KCC) and the efflux of organic osmolytes [1,2,3,4,5,6,7,8].

There are surprising multiple and diverse signaling molecules participating in signal transduction from volume sensor(s) onto executive ion transport proteins. Serine/threonine kinases and phosphatases, protein tyrosine kinases, mitogen-activated protein kinases (MAPKs), small GTPases of the Rho family, reactive oxygen species (ROS) participate in shrinkage- and swelling-induced signaling [2,7,9,10].

Among members of the superfamily of myosin motors we only discuss non-muscle myosin II (NMMII). Experimental data on involvement of NMMII in cell volume regulation are rather scarce, but it has been reported that cytoplasts from Ehrlich ascites tumor cells lack both, NMMII and shrinkage-induced activation of the NKCC1 [11]. Phosphorylation of myosin light chain by myosin light chain kinase (MLCK) was elevated and decreased under cell shrinkage and swelling, respectively, in endothelial cells [12]. The basal activity of myosin ATPase appears to be permissive for activation of NKCC by cell shrinkage [13,14].

Here, we discuss the complex consisting of the plasma membrane (PM) and the submembrane actin-based cytoskeleton (smACSK), which we shall refer to as PM—smACSK. We regard PM—smACSK as a viscoelastic solid with four effects on active ATP-consuming cell volume sensing: firstly, NMMII-generated pulling force directed into inside of the cell. Secondly, a force of protrusive actin polymerization within the PM—smACSK would generate a pushing force directed out from the cell. Thirdly, PM—smACSK would be a rigid structure that should be pulled in by NMMII or pushed out by F-actin assembly. Fourthly, PM-smACSK is considered as being a spring. Indeed, spring-like behavior seems to be necessary for a sensor of changes in membrane curvature to function because it must return to its baseline conformation upon release of the stimulus, i.e., when the cell restores its normal volume and shape. Morishima et al. have shown that the swelling-induced activation of the Cl^−^ channel depends on the spring energy of the membrane [15].

We hypothesize that NMMII and force of actin-based protrusions can probe balance of hydrostatic pressures (HPs) across the plasma membrane as well as changes in ionic strength in submembrane compartments. To function as sensors of the cell volume NMMII and the actin-based motor should meet at least four requirements: (i) to be controlled by the same signaling pathways that determine cell fate (discussed in Section 2), (ii) to unambiguously distinguish between swelling and shrinkage (discussed in Section 3), (iii) to sense changes in the intracellular ionic strength (discussed in Section 4), and (iv) to possess mechanisms of transduction of information from NMMII and F-actin to signaling pathways mediating RVI and RVD (discussed in Section 5).

## 2. Tuning of Cell Volume Set Point by Cell Program (Proliferation, Apoptosis, Migration) via Control of NMMII, F-Actin Protrusive Force and PM-smACSK

Cell fate determines the whole architecture and biomechanics of the CSK. Here we subdivide the whole CSK into the bulk CSK, spanning through the cell interior, and the submembrane CSK which underlines the PM. Stimuli that switch on a cell program (e.g., proliferation, migration, apoptosis) activate multiple signaling pathways that regulate the assembly of the actin-based CSK, and the architecture of the smACSK determines the PM-smACSK biomechanics and cell volume sensing (Figure 1).

### 2.1. NMMII, a Pulling Motor

#### 2.1.1. NMMII, Common Features and Regulation

NMMII is an actin-based molecular motor. It is a heterohexamer comprised of two heavy chains (HCs), two essential light chains (ELCs) and two regulatory light chains (RLCs) [16,17,18,19]. Each heavy chain consists of 3 domains: the N-terminal globular head harboring the ATPase activity responsible for force generation and actin binding; the neck domain which binds to ELCs and RLCs, and the C-terminal α-helical rod domain with a nonhelical tail-piece needed for thick filament formation and cargo binding [16,17,18,19]. This heterohexameric complex of 2 HCs, 2 ELCs and 2 RLCs is referred to as NMMII monomer, since it represents a functional unit, and NMMII monomers undergo assembly into filaments and stacks [19] (Figure 2). Depending on their HCs there are three isoforms of NMMII (NMMIIA, NMMIIB and NMMIIC) that differ in tissue distribution, function, intracellular localization and regulation [16,17,18,19].

The actin-activated ATPase activity of the NMMII heavy chain is stimulated by phosphorylation of Ser19/Thr18 of RLC by Ca^2+^/calmodulin-dependent MLCK [20]. Myosin light chain phosphatase (MLCP), where serine/threonine phosphatase 1 (PP1) acts as catalytic subunit, inhibits NMMII via dephosphorylation of Ser19/Thr18 in RLC [21,22,23]. The small GTPase RhoA stimulates its target ROCK (Rho-associated coiled-coil containing kinase) which activates NMMII contractility both via phosphorylation of Ser19/Thr18 of RLC [24] and phosphorylation and inhibition of MLCP [25]. MLCK can be inhibited via phosphorylation by p21-activated kinase (PAK) [26]. In PC12 and N1E-115 cells, Rac1 was shown to induce phosphorylation and inhibition of NMMII in PAK1-independent, Ca^2+^-dependent manner [27]. Phosphorylation/dephosphorylation of heavy chains of NMMII also control behavior of NMMII [17].

#### 2.1.2. Evidence for Oscillatory Activity of NMMII

Sensing and regulation of cell volume is of critical importance for cell survival or induction of proliferation, apoptosis or migration. We suggest that NMMII is constantly measuring the gradient of HP across the elastic PM-smACSK complex. A possible role for NMMII in measuring of HP_in_ (intracellular HP) and HP_out_ (extracellular HP) would be repetitive cycles of NMMII activity: contraction of NMMII followed by relaxation as it is described in Section 3. It is therefore of interest to ascertain whether oscillatory activity of NMMII takes place in cells. In mouse embryonic fibroblasts oscillatory activity of NMMII localized at the lamellipodium participates in formation of integrin-mediated adhesion sites [28]. In a number of cell types, mouse embryonic fibroblasts, human breast adenocarcinoma MCF-7 cells and human osteosarcoma U2OS cells, the dynamics of NMMII activity were studied [29]. The authors revealed a localized pulsatile activity of the NMMIIA isoform, while NMMIIB did not show pulsing behavior [29]. The oscillatory activity of NMMIIA was dependent on Ca^2+^ influx and the activation of MLCK [29].

#### 2.1.3. Can NMMII Reset Its Own Oscillatory Activity?

It would be important to understand whether the oscillatory activity of NMMII (discussed above) can be driven by NMMII itself. The RhoA/ROCK module can activate NMMII [24,25], while Rac1/Pak can inhibit it [26,27] (Figure 3). RhoA and Rac1 are themselves under the control of guanine nucleotide exchange factors (GEFs) [30]. All GEFs possess two tandem domains in their catalytic core: DH [Dbl (diffuse B-cell lymphoma) homology] domain and a PH (pleckstrin homology) domain. The largest family of GEFs is the Dbl family [30,31,32]. NMMIIA and NMMIIB directly bind to multiple GEFs of the Dbl family: Trio, GEF-H1, FGD1, Kalirin, βPIX, LARG, Dbl, Tiam1, Vav1 [33]. DH-PH domains of Dbl family GEFs interact with the head domain of NMMIIB [33]. Assembly of NMMII into filaments and ATPase activity are required for binding to Dbl GEFs. When bound to NMMII, GEFs are inactive. NMMII inhibition with blebbistatin released Dbl GEFs from NMMII and resulted in their reactivation [33]. βPIX is a Rac1/Cdc42 specific GEF [33] and its activation may activate Rac1-PAK axis [34] that can inhibit NMMII [26,27]. Surprisingly, when inhibited, NMMII releases and activates βPIX and other Dbl family GEFs [33], that might further inhibit NMMII activity (Figure 3). In case of release of active RhoA-specific GEF-H1 from NMMII [33] one may expect activation of the RhoA/ROCK axis and activation of NMMII that would complete the self-supporting cycling activity of NMMII (Figure 3). However, experimental data show dissociation of GEF-H1 from NMMII, but not activation of GEF-H1 [33]. In addition, Rac can downregulate Rho [35,36]. GEF interacting with NMMIIA was identified in HeLa cells and termed MyoGEF (myosin-interacting GEF) [37]. MyoGEF is inactive when bound to NMMIIA and is activated upon release from NMMIIA; MyoGEF activates RhoA but not Rac1 [37].

Thus, active NMMII is likely to inhibit its own activity via the βPIX–Rac1–PAK pathway [33]. Though the state of NMMII ATPase activity was not determined, NMMII could also activate NMMII itself via the MyoGEF–RhoA–ROCK pathway [37].

In summary, the experimental data are not conclusive to underpin the proposed self-supporting oscillatory model for NMMII (Figure 3). Alternatively, NMMIIA can drive deformations of the PM, opening of mechanosensitive Ca^2+^-permeable cation channels, Ca^2+^ influx, activation of MLCK and stimulation of NMMIIA [29].

### 2.2. A Viscoelactic Solid Composed of the PM and the smACSK, a Pushing Motor, a Resistive Force and a Spring

We consider here the three mechanical properties of the PM-smACSK: as a pushing motor, as a resistance to pushing/pulling and as a spring. However, all four mechanical parameters, NMMII is also a constituent of the smACSK, are characterizing the same viscoelastic solid composed of hundreds of proteins that hinders clear distinction of mechanism of generation of any one force.

#### 2.2.1. The PM-smACSK Complex, Common Features

Here we define the PM as a lipid bilayer with transmembrane proteins. The lipid bilayer as a solid can be characterized by four elasticity moduli: a compressibility modulus, an area expansion modulus, a bending modulus and an elastic shear modulus [38]. Inclusions, e.g., eicosapentaenoic acid and docosahexaenoic acid, in the lipid bilayer can change its elasticity [39]. The presence of molecular dioxygen in the lipid bilayer is also likely to influence its mechanical properties and the activities of transmembrane signaling proteins [40].

Underlining the PM is the smACSK composed mainly of spectrin and actin filaments [41,42]. The spectrin–F-actin cytoskeleton is particularly well studied in red blood cells [43,44]. Auxiliary proteins such as ankyrin, protein 4.1R, adducin, dematin, tropomyosin and tropomodulin provide “vertical” (between the lipid bilayer and smACSK, directed perpendicularly to the plane of the PM) and “horizontal” (between the constituents of the smACSK, directed tangentially to the plane of the PM) linkages [43,44]. Ankyrin is mainly responsible for connection of Band 3 (anion exchanger 1) with the spectrin—F-actin network, while protein 4.1R brings together glycophorin C and submembrane CSK [43,44]. The phosphorylation status of Band 3 influences the erythrocyte deformability [45,46,47,48,49].

In addition, proteins of the plakin family link three major constituents of the CSK F-actin, microtubules and intermediate filaments together [50]. In particular, microtubule actin crosslinking factor 1 [MACF1, also known as actin crosslinking family 7 (ACF7)] integrates actin filaments and microtubules [51]. This MACF1 mediated connection between F-actin and microtubules may serve for signal transduction onto and from microtubules for which we have recently hypothesized a role in the integration of intracellular signaling events [52].

#### 2.2.2. Protrusive F-Actin Polymerization, a Pushing Motor

Actin polymerization drives cell motility and migration [53,54,55]. We propose that the pushing force generated by protrusive actin filaments [54] would probe the strength of HP_out_ that drives the PM-smACSK into the cell in case of cell shrinkage. Assembly of protrusive F-actin may occur as formation of filopodia, finger-like protrusions with parallel actin bundles, or lamellipodia, flat protrusions formed by branched actin filaments [54].

#### 2.2.3. A Resistive Force Generated by the PM-smACSK

When the PM-smACSK complex is being pulled into cell by NMMII, the NMMII contractility is to overcome the rigidity of the PM-smACSK. The rigidity of the PM-smACSK is mainly determined by elasticity of proteins comprising the smACSK and by the strength and number of “vertical” and “horizontal” links between the constituents of the PM-smACSK. The force acting against movement of the PM out from cell is referred to as membrane tension, a complex parameter including the in-plane tension in the lipid bilayer and the energy of adhesion between the PM and the smACSK [56]. However, in fish keratocytes the force generated by actin protrusions can increase membrane tension [56].

#### 2.2.4. Spring-Like Behavior of the PM—smACSK

According to our hypothesis the complex composed of the PM with tightly bound smACSK undergoes NMMII- and F-actin-driven cycles of displacements in directions normal to the lipid bilayer (Figure 4). Deformation of the lipid bilayer and the underlining CSK can vary from completely elastic deformation (Figure 4, upper panel) to completely plastic deformation (Figure 4, lower panel). In the case of elastic deformation, the PM-smACSK completely recovers its shape, while in the case of plastic deformation the PM-smACSK retains change in its form (Figure 4).

### 2.3. Cell Fates Requiring Volume Change

#### 2.3.1. Proliferation

Cell proliferation requires a shift in the cell volume set point to a greater cell volume [1,2]. In our hypothesis cell program-dependent changes in four mechanisms are necessary for adjustment of the volume set point: NMMII activity, F-actin protrusions, resistive and elastic behavior of the PM-smACSK complex. EGF can trigger both proliferation and migration [57,58,59], but the mechanism discriminating between EGF-induced proliferation and migration yet remains elusive. The EGF receptor family comprises four members ErbB1/HER1, ErbB2/HER2/Neu, ErbB3/HER3 and ErbB4/HER4 [59]. The three main signaling pathways activated by EGF are Ras-Raf-MEK1/2-ERK1/2, Akt-PI3K-mTOR and PLC-γ1-PKC [59]. Ras, a family member of the small GTPases downstream of growth factor receptors [60], via Rho—ROCK pathway may thus control both NMMII [61] and actin CSK [62,63].

In a kidney proximal tubule epithelial LLC-PK1 cell line and a canine distal tubular epithelial MDCKII cell line, EGF activates RhoA via ERK-mediated activation of GEF-H1 [64]. In COS-7 cells, EGF was shown to activate ERK which further phosphorylated RhoA at Ser-88 and Thr-100 that enhanced RhoA activity, increased interaction between RhoA and its target ROCK1, ROCK1 phosphorylated MYPT1 [58]. EGF-induced ERK-mediated RhoA phosphorylation increased stress fiber formation [58]. In COS-7 cells, EGF was shown to activate Rac1 [57]. In COS-7 cells EGF activated ERK and ERK phosphorylated Rac1 at Thr-108 that decreased Rac1 activity and targeted Rac1 to the nucleus [57].

Since we propose that EGF tunes mechanosensitive mechanisms in the cell, the recent observation on fine-tuning of mechanics of integrin by EGF is of interest [65]. In Cos-7 cells, EGF attenuates the threshold of integrin tension generation and increases focal adhesion assembly and cell spreading [65].

GPCRs are among key drivers of cancer cell growth [66], they are also involved in cell volume regulation [67]. Via β-arrestins and small GTPases, GPCRs can regulate filamin-A, cortactin, cofilin and thus regulate re-arrangements of the CSK that contribute to cancer cell survival and proliferation [66]. In the context of our hypothesis GPCR—the β-arrestin axis via effects on the CSK [66] may regulate both rigidity of the CSK and its spring-like behavior, i.e., the ability of the submembrane spectrin—F-actin CSK to store energy of deformation.

#### 2.3.2. Migration

Migration of a single cell on a 2D substrate can be divided into two modes: mesenchymal and ameboid [68]. Mesenchymal migration occurs via 5 successive steps: (1) formation membrane protrusions at the leading edge; (2) assembly of focal adhesions; (3) proteolysis of the extracellular matrix by cell surface proteases; (4) translocation of the cell body; and (5) retraction of the rear of the cell [69]. Localized swelling and shrinkage take place at the leading edge and trailing edge, respectively, that requires concerted work of multiple ion channels and transporters [70]. In our model, re-modeling of both submembrane CSK and changes in activities of NMMII would tune volume set points, that appear to be different at the front and the rear of the migrating cell.

A number of signaling proteins regulate cell migration including the small GTPases Rho, Cdc42 and Rac [62,71,72]. The Rho-ROCK axis governs both NMMII activity [24,25] and actin-based CSK architecture [62,71,72]. In the prostate carcinoma TSU-pr1 cell line, EGF can induce chemotaxis and was shown to regulate NMMIIA and NMMIIB heavy chain phosphorylation by PKC [73]. It should be noted that in migrating cells, NMMIIA and NMMIIB seem to be regulated in distinct modes [73] and to be asymmetrically re-distributed [74,75,76]. NMMIIA was shown to move to the leading edge, while NMMIIB was shown to accumulate at the rear of the cell [74,75,76]. It appears that the C-terminal tails of heavy chains of NMMIIA and NMMIIB determine differential distribution of these isoforms in migrating cells [76]. Moreover, in addition to their motor activities, NMMIIA and NMMIIB can participate in the control of membrane protrusion formation and dynamics of focal adhesions [77].

#### 2.3.3. Apoptosis

Execution of apoptosis depends on a decrease in cell volume, that was termed apoptotic volume decrease (AVD) [6,8]. In our hypothesis, AVD may be linked to a NMMII- and PM-smACSK-dependent shift in volume set point, since signaling pathways mediating cell death can also control NMMII. In PC12 cells and cultured rat cortical neurons oxidative stress-induced apoptosis involves sequential activation of caspase-3, ROCK1 and myosin light chain (MLC) that leads to NMMIIA activation [78]. Inhibition of NMMIIA with blebbistatin was shown to suppress cell death in human red blood cells [79], in neurons [78], in hair cell-like HEI-OC-1 cells and in cochlear hair cells [80].

Voltage-dependent anion-selective channel 1 (VDAC1), originally described as a mitochondrial outer membrane protein [81], is also expressed in the plasma membrane [82], where it enriches in caveolae [83] and lipid rafts [84] and has been implicated in cell volume control [85]. VDAC1 is known to directly bind to actin [86,87]. In neuronal cells, opening of plasma membrane VDAC1 channels was described to precede caspase activation and antibodies against VDAC1 were able to prevent apoptosis in these cells [88].

## 3. “Two Motors and One Spring” Model

Our two motors and one spring model consists of three core elements: NMMII, as a pulling motor, F-actin-based protrusions as a pushing motor, and PM-smACSK as a spring. These three elements can act together in measuring the strength of the hydrostatic pressure that does not displace the PM—smACSK under normal cell volume (Figure 5A) but drives inward displacements of the PM—smACSK during cell shrinkage (Figure 5B) and outward displacements during swelling (Figure 5C). It should be mentioned that the plasma membrane in living cells is ever undergoing the excursions in directions perpendicular to the plane of the membrane, the phenomenon known as cell membrane fluctuations [55,56]. It would be tempting to assume that NMMII drives cell membrane fluctuations, however, at least in human red blood cells, NMMII was shown not to contribute to membrane fluctuations [57].

When a cell is in an isosmotic milieu the intracellular hydrostatic pressure (HP_in_) is equal to the extracellular hydrostatic pressure (HP_out_) (Figure 5A). Exposure of a cell to a hyperosmotic solution induces an efflux of water resulting in an increase of the hydrostatic pressure in the surrounding milieu, HP_out_, and cellular compression: the cell shrinks (Figure 5B). Exposure of the cell to an hypoosmotic solution induces influx of water resulting in an increase of the intracellular HP_in_ and inflation of the cell: the cell swells (Figure 5C).

PR65, a scaffolding subunit of protein phosphatase 2A (PP2A), possesses a HEAT-repeat (Huntingtin, elongation factor 3, a subunit of protein phosphatase 2, PI3 kinase target of rapamycin 1) [89]. Studies of the elastic behavior of the HEAT-repeat protein PR65 show that extension of PR65 opens the substrate/catalysis interface and activate PP2A, while compression of PR65 closes the substrate/catalysis interface and inhibits PP2A [89]. Thus, PR65 can link mechanical stress sensing and activation/inhibition of the signaling protein PP2A. In Calu-3 airway, epithelial cells F-actin serves as a scaffold for bringing together NKCC1 and its regulators such as protein kinase C-δ (PKC-δ), STE20-related proline-alanine-rich kinase (SPAK) and PP2A [90]. PP2A dephosphorylates and inhibits NKCC1 with which was co-immunoprecipitated from Calu-3 cell lysates [90]. In addition, PP2A can dephosphorylate and activate the KCC [91].

Considering the behavior of any mechanosensitive transmembrane or membrane-associated protein, e.g., PP2A [89], under cell shrinkage and swelling in an oversimplified approximation (Figure 6). For a mechanosensitive protein to be activated/inhibited by mechanical stress it needs to be fixed at least at two points, e.g., at the plasma membrane and the bulk microtubule-based CSK (Figure 6A). We assume the bulk microtubule-based CSK as being immobile during cell volume changes that allows us to use it as a reference point for the assessment of the PM-smACSK displacement and of degrees of compression/distension of any single protein in radial direction (Figure 6B). The same compressive force (HP_out_) that changes membrane curvature under cell shrinkage would compress any mechanosensitive protein in the cell, while increase in HP_in_ during cell swelling would distend it (Figure 6C).

If one excises any small patch of the PM-smACSK from a cell with normal volume, then this patch can be regarded as planar (Figure 6 and Figure 7). In our model any local planar patch of the PM-smACSK (i) undergoes NMMII-dependent outward excursion, (ii) then NMMII relaxes and the PM-smACSK spring returns the PM-smACSK into planar shape, (iii) activation of NMMII pulls the PM-smACSK into the cell, and (iv) relaxation of NMMII and the PM-smACSK spring returns the PM-smACSK in planar shape (Figure 7). Signal transduction from NMMII and F-actin onto signaling proteins is discussed in Section 5.

### 3.1. Hyperosmolarity-Driven Inward Movement of the PM—smACSK versus F-Actin-Driven Outward Movement of the PM—smACSK: Sensing of Cell Shrinkage

Assuming that in order to detect the increase in HP_out_—that pushes the PM-smACSK into the cell interior—the cell would generate a force that drives the PM-smACSK in the opposite direction (Figure 7A–C). In our model, PP2A is tethered to both PM and smACSK (Figure 8). Cell shrinkage compresses and inactivates PP2A as observation on PP2A mechanosensitivity allows to suggest [89] (Figure 9). This inhibition of PP2A leads to phosphorylation and activation of NKCC1 [90] and to phosphorylation of KCC that inhibits KCC during shrinkage [91] (Figure 9).

### 3.2. Relaxation of Both F-Actin-Driven Force and the PM—smACSK Spring

We presume the PM-smACSK is an elastic solid assuming dome-like shape under NMMII-dependent excursion of the PM-smACSK. Being an elastic solid the PM-smACSK stores energy of deformation and, as soon as NMMII is inactive, the elastic PM-smACSK returns to its planar shape (Figure 7C,D). Also, this relieve of PM-smACSK tension returns the PP2A to its normal state.

### 3.3. Hyposmolarity-Driven Outward Movement of the PM—smACSK versus NMMII-Driven Inward Movement of the PM—smACSK: Sensing of Cell Swelling

The next step in this hypothetical cycle of NMMII activity is the activation of NMMII and pulling the PM-smACSK into the cell against the increased HP_in_ that drives the PM-smACSK out of the cell in case of cell swelling (Figure 7D–F).

In our model, cell swelling extends and thus activates PP2A as the study of PP2A mechanosensitivity suggests [89] (Figure 10). Activation of PP2A leads to dephosphorylation and inactivation of NKCC1 [90] and to dephosphorylation and activation of KCC [91] during cell swelling (Figure 10). Some other connections between NMMII and signaling proteins, involved in RVD, are discussed in Section 5.

### 3.4. Relaxation of the Both NMMII and the PM—smACSK Spring

Inward displacement of the PM-smACSK, that is transition of the PM-smACSK from planar to dome-like shape, results in an accumulation of energy of deformation in the PM-smACSK. Relaxation of NMMII relieves a force acting on the PM-smACSK and the PM-smACSK returns from dome-like into planar shape (Figure 7F). Thus, one hypothetical cycle of concerted work of NMMII and PM-smACSK spring is completed and the next cycle of NMMII and PM-smACSK spring dynamics can begin (Figure 7).

## 4. How Can NMMII, and the PM—smACSK Complex Sense Intracellular Ionic Strength?

Salt bridges and hydrogen bonds between oppositely charged residues may underlie sensing of intracellular ionic strength by NMMII and F-actin protrusions.

Non-muscle myosin was identified in human erythrocytes [92]. Later on, human erythrocytes were shown to express the NMMIIA isoform [93]. ATPase activity of human erythrocyte NMMII is very low at low ionic strength (0.06 M KCl) in the presence of EDTA, a calcium chelator, or in the presence of MgCl_2_ [92]. Moreover, NMMII appears to distinguish between Na^+^ and K^+^ cations, since ATPase activity of NMMII in the presence of 2 mM EDTA was reported as 0.376 μmol/min per mg in 0.5 M KCl, but as 0.118 μmol/min per mg in 0.5 M NaCl [92]. Similarly, the ATPase activity of NMMII in the presence of 10 mM CaCl_2_ was about 2-fold higher (0.478 μmol/min per mg) in 0.5 M KCl than in 0.5 M NaCl (0.204 μmol/min per mg) [92]. The presence of negatively and positively charged regions in the coiled-coil rod domains of NMMII [94,95] may be responsible for its sensitivity to ionic strength. F-actin itself is negatively charged and can bind to positively charged lipids [96]. Numerous proteins that provide assembly of the actin CSK possess negatively and positively charged domains involved in protein-protein interactions. Palladin, an actin cross-linking protein, possesses two basic sequences on opposite sites of an immunoglobulin 3 domain which provide electrostatic interaction of paladin with two actin filaments [97].

Negatively charged G-actin and F-actin can bind to positively charged lipids [96]. Positively charged pleckstrin homology (PH) domain in various adaptor and signaling proteins provide binding of these proteins to negatively charged phosphoinositides [98].

## 5. NMMII Can Directly Interact with and Modulate Activities of Transmembrane Mechanosensors of the PM

### 5.1. Anionic Lipids

Pulling or pushing of the lipids of the PM can generate mechanical pressure in the lipid bilayer that would be sensed by transmembrane mechanosensors [38,99,100]. Actin CSK can interact with the membrane phosphoinositides via PH domains of adaptor proteins [98,101].

C-terminal fragments of heavy chains of NMMII bind directly to phosphatidylserine (PS) lipid vesicles and liposomes [102,103,104,105] and this binding is independent of F-actin [105]. It was demonstrated that N-terminal myosin heads could also interact with membrane phosphatidylserines [105]. Interestingly, binding of NMMII to acidic liposomes leads to displacement of RLCs from the myosin heavy chains [105] and an excess of RLCs suppressed its binding to phosphatidylserine vesicles suggesting phosphorylation of RLCs may regulate the interaction of NMMII to PS and downstream signaling. In *Drosophila* neural stem cells NMMII RLC, Sqh (Spaghetti-squash), binds to a number of phosphoinositides: a phosphatidylinositol 4-phosphate [PtdIns(4)P], phosphatidylinositol 3,4-bisphosphate [PtdIns(4,5)P2], phosphatidylinositol 3,4,5-trisphosphate [PtdIns(3,4,5)P3], phosphatidylethanolamine, and is anchored to cell cortex by phosphatidylinositol transfer protein Vibrator [106]. This sqh, PIs, and vibrator complex regulates asymmetric cell division.

### 5.2. Ion Channels

Ion channels are pore-forming membrane proteins located in cellular membranes and allow the passage of ions through the channel pore [107]. NMMII seems to interact with several membrane localized ion channels where it plays important role in channel trafficking and spatial localization. For example, in rat embryonic neurons NMMIIB binds to Kv2.1 ion channel via its C- and N-termini and regulates its transport from cytoplasm to membranes [108]. Pharmacological inhibition or knockdown of NMMIIB both suppress Kv2.1 membrane translocation [108]. Likewise, it has been shown that various NMMII isoforms directly interact with Na^+^/K^+^-ATPase α1 subunit and play a role in its transport and trafficking both in neuronal and non-neuronal cells [109]. Furthermore, NMMIIB in neurons interacts with Cav2.1 Ca^2+^ channel and is involved in its localization to membrane [110]. Alternatively, ion-channel activity may also regulate NMMII activity. For example, in megakaryocytes, TRPM7-mediated Mg^2+^ homeostasis regulates NMMIIA activity that regulates platelet generation from megakaryocytes [111]. Direct interaction between actin CSK and mechanosensitive ion channels are reviewed elsewhere [38,99].

### 5.3. Integrins

Integrins are heterodimeric adhesion molecules consisting of an α and a β subunit and link the intracellular signaling components and actin cytoskeleton to the extracellular matrix. They are structural proteins playing an important role in mechanosensory transduction and they mediate both inside-out and outside-in signaling [112]. Integrins are tethered to the actin CSK via a number of adaptor proteins, e.g., talin, vinculin, α-actinin, filamin, tensin, kindlin [112]. Integrins’ role as mechanosensors can proceed via three levels: first level, modulation of integrin conformation by mechanical force; second level, mechanosensitivity of the integrin-ligand binding; third level, mechanosensitivity of integrin-linked proteins, e.g., talin [112].

Several lines of evidence suggest that, at least in hepatocytes, they act as volume sensors and are clustered in response to cell volume changes [113,114,115,116]. Clustering of integrins in response to volume changes mediates signaling via their cytoplasmic tails to various intracellular pathways leading to activation of ion channels involved in the adjustment of regulatory cell volume [116,117,118,119,120].

Interaction between integrins and NMMII is less studied field. The Stouffer lab first reported accumulation of NMMII heavy chains at alpha v beta 3 integrin complexes in response to vascular injury in smooth muscle cells [121]. Later Rivera Rosado et al. demonstrated a physical interaction between the NMMIIA heavy chain and the alpha 4 integrin tail that plays a role in cell migration [122]. Likewise, the NMMIIA heavy chain is recruited to integrin lymphocyte function-associated antigen (LFA)-1 where it regulates lymphocyte tail detachment and migration [123]. Focal adhesion kinase (FAK), a non-receptor protein tyrosine kinase, is involved in both RVI and RVD [2]. FAK can bind to growth factor receptor bound protein 2 (Grb2) [124], while Grb2 appears to directly interact with NMMII [125].

### 5.4. G protein Coupled Receptors (GPCRs)

It is well documented that activation of various GPCRs can influence the activity of NMMII via activation of downstream kinases and phosphatases and thus may play a role in cell-volume regulation [61]. However, data on direct interaction of NMMII and GPCRs is scarce. A recent study has highlighted an interaction between CXCR4 and NMMIIA [126]. CXCR4 is a GPC chemokine receptor that shuttles between the plasma membrane and nucleus [127]. Xu and colleagues have reported that NMMIIA interacts directly with CXCR4 mediating its nuclear translocation and cell migration [126]. In HEK-293 cells actin CSK destabilization with cytochalasin D was shown to influence signal transduction through the serotonin_1A_ receptor [128].

### 5.5. Growth Factor Receptors or Receptor Tyrosine Kinases (RTKs)

There is the surprising observation that epidermal growth factor receptor (EGFR) can be activated by both swelling and shrinkage [129]. Downstream of RTKs are phosphoinositide 3 kinase (PI3K), 3-phosphoinositide-dependent protein kinase-1 (PDK1) and protein kinase B (PKB, also known as Akt) [130]. In Swiss 3T3 fibroblasts, hyposmolarity activates the EGFR/PI3K/Akt axis, resulting in taurine efflux [131]. The muscle isoform of myosin II isolated from rabbit skeletal muscle binds to the pleckstrin homology (PH) domain of PKB [132]. Interaction of the PH domain of PKB with PtdIns(3,4,5)P3 and PtdIns(4,5)P2 produced by PI3K induces conformational change in PKB making PKB activatable by PDK1 [133]. Questions arise if NMMII can bind to the PH domain of PKB, and if contractile activity of NMMII can influence the PKB conformational change responsible for activation of PKB by PDK1.

Tyrosine-phosphorylated EGFRs and platelet-derived growth factor receptors (PDGFRs) can directly interact with growth factor receptor bound protein 2 (Grb2) which transmits the signal further onto the small GTPase Ras [134]. In NIH3T3 cells Grb2 appears to directly interact with dynamin II [135] and NMMII [125]. Also, dynamin II seems to directly interact with NMMII in NIH3T3 cells [125]. This association of RTKs with NMMII mediated by Grb2 and dynamin II [125,135] might transmit a traction force generated by NMMII onto RTKs and induce their activation under cell swelling. Grb2, which was reported to associate with NMMII [125], can participate in signal transduction from a NMMII generated force onto Ras [134]. Ras itself is involved in activation of mitogen-activated protein kinases (MAPKs) [136], while MAPKs are well known to mediate cell responses to osmotic challenges [2]. Dimerization of the EGFR depends on remodeling of the actin CSK [137].

### 5.6. Angiotensin-Converting Enzyme (ACE)

ACE is a membrane protease expressed in several tissues and an important component of the renin-angiotensin system (RAS). It converts angiotensin (Ang) I to Ang II which regulates blood pressure through its effects on the kidneys, brain, heart and blood vessels. In addition to the cleavage of Ang I to Ang II, ACE has been reported to interact directly with NMMII and participate in outside-in signaling in endothelial cells [138]. Based on these data, we propose ACE may be involved in cell-volume regulation via its direct interaction with NMMII.

## 6. Towards the Experimental Testing of the Hypothesis in Live Cell in Real Time

Experimental testing of our model requires the real-time live-cell observation of the cascade of the following signaling and executive events.

Visualization of activation of cell surface receptors—that is observation of signaling that fine-tune the NMMII and smACSK;Visualization of signaling proteins that transmit signal from cell surface receptors onto NMMII and smACSK;Observation of activation of NMMII and re-arrangements of the smACSK;Recording of pulling force and pushing force exerted by NMMII and actin protrusions, respectively;Visualization of activation of transmembrane signaling proteins by NMMII-driven pulling force and actin protrusion-driven pushing force;Observation of activation of signaling proteins that transduce signal from the transmembrane signaling proteins onto ion transport systems;Observation of activation of ion transport systems responsible for RVI and RVD.

In our model cell surface receptors are to be activated twice, when they adjust cell volume point to execution of particular cell program, and when they are activated by mechanical stresses generated by cell swelling, shrinkage and NMMII and protrusive F-actin. Thus, signaling from EGFR is suggested to tune volume sensing machinery (see Section 2.3) and EGFR itself can be activated both by swelling and shrinkage [129]. <EGFR visualization>. Swelling of neuronal cells occurs under pathological conditions such as hyponatremia and hypoxia/ischemia [139]. Neuronal cells respond to swelling via efflux of inorganic and organic osmolytes (Fisher et al., 2008+). Ligand-dependent activation of various GPCRs contributes to stimulation of osmolytes efflux in neuronal cells (Fisher et al., 2008+). Methods for real-time live-cell observations of GPCRs’ activation are available [140,141]. Activation of Gq protein can also be visualized in live cell in real time [142].

Protein tyrosine kinases of the Src family, presented by nine members (Src, Fyn, Yes, Fgr, Blk, Lck, Hck, Lyn and Yrk), participate in both RVI and RVD [143]. Real-time live-cell visualization of Src and Fyn, activation can be achieved [144,145]. <NMMII and smACSK visualization>. To evaluate the forces acting on any singles protein, the molecular tension sensors are being developed [146,147]. As concerns live-cell measurement of ion transport system, activity of NHE1 can be monitored with cSNARF1, a fluorescent sensor of intracellular pH [148].

## 7. Concluding Remarks

Tight connections between cell volume and cell fate highlight the importance of the search for the molecular apparatus that performs cell volume sensing and regulation. NMMII and F-actin assembly acting together with a PM—smACSK spring seem a promising candidate volume sensor to us. We present an hypothesis that comprises four key features, (1) fine-tuning of cell volume set point (i.e., NMMII, F-actin protrusive activity and viscoelastic properties of PM-SMACSK complex) by cell fate, (2) probing of changes in difference in hydrostatic pressures (HP_in_ versus HP_out_) across the PM by two active ATP-consuming mechanisms (NMMII and F-actin polymerization), (3) perfect elasticity of the PM-SMACSK complex, allowing its return to normal shape, and (4) sensing of changes in intracellular ionic strength by NMMII and machinery performing F-actin polymerization. Direct associations of NMMII and F-actin with several transmembrane signaling molecules provide the route for signal transduction onto ion transporting systems.

The search for any particular volume-sensitive—or, as we assume here, active, energy-consuming cell volume sensing—molecular apparatus would be of importance for, e.g., anti-cancer treatment. Uncontrollable cell proliferation, resistance to apoptosis, migration—these features are among the hallmarks of cancer cell biology [149]. These cancer cell activities are critically dependent on volume regulatory systems, in particular on shifts in the volume set point. Ion channels are involved in the regulation of cell proliferation and apoptosis as well as in cell volume regulation [1,4,6,7,8,9]. NMMII and PM-smACSK could be a signaling hub linking execution of cell proliferation, migration and apoptosis to signaling pathways controlling activities of ion channels and transporters. NMMII isoforms appear to regulate the metastatic potential of cancer cells and, thus, to be novel therapeutic targets [150,151,152]. Our hypothesis on NMMII as a key player in cell volume sensing broadens the spectrum of effects of NMMII in cancer cells. (Patho)physiological roles of NHEs in organ systems—cardiovascular system, brain, kidneys, skeletal muscle, gastrointestinal tract, salivary glands, pancreas, eyes, immune system, reproductive system, bone, skin—were recently reviewed [9]. Our model suggests a mechanism for activation of NHE isoforms by cell shrinkage and swelling.

## Figures and Tables

**Figure 1 ijms-22-07967-f001:**
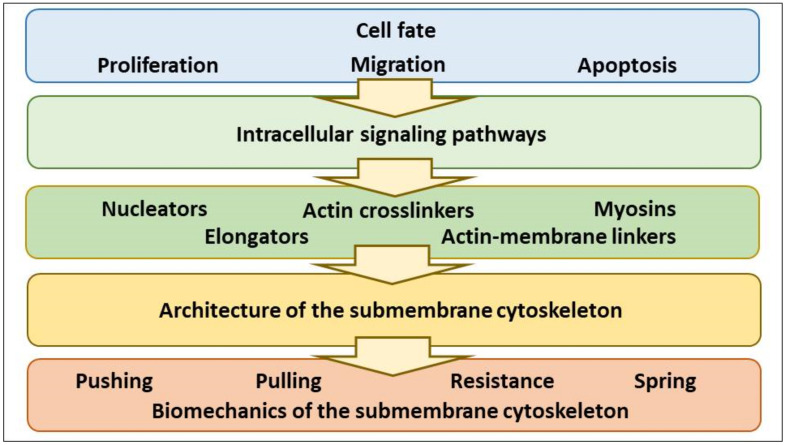
The schematizing of signal transduction from the cell fate-controlling stimuli onto the biomechanics of the PM-smACSK complex.

**Figure 2 ijms-22-07967-f002:**
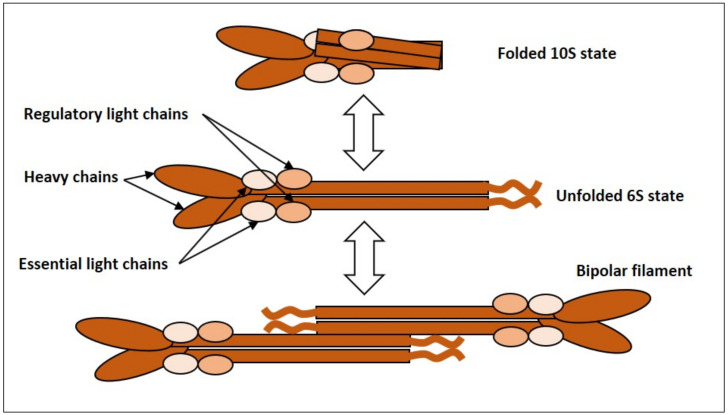
NMMII monomer refers to the heterohexameric complex composed of 2 heavy chains, 2 essential light chains and 2 regulatory light chains. NMMII can exist in a folded assembly-incompetent 10S state (top), and an unfolded assembly-competent 6S state (middle). The 6S NMMII can assemble into bipolar filaments (bottom).

**Figure 3 ijms-22-07967-f003:**
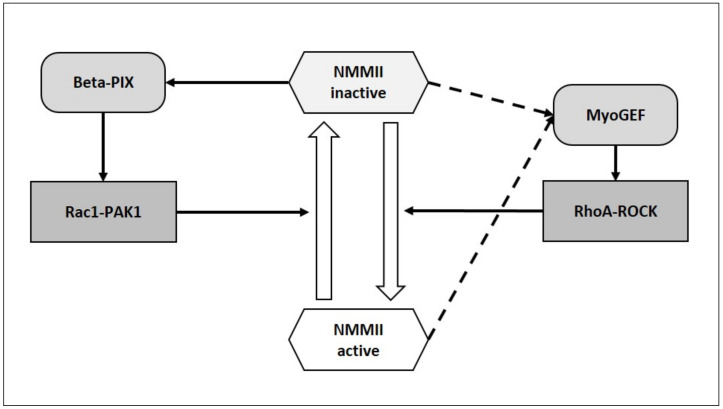
NMMII might induce positive feedback loops. When ATPase activity of NMMII is inhibited, it can release GEFs of the Dbl family, which are likely to further inhibit NMMII via activation of the Rac1-PAK pathway. It is not clear which state of NMMII (active or inactive ATPase activity) can induce the release of active MyoGEF. Active MyoGEF can stimulate NMMII via activation of ROCK.

**Figure 4 ijms-22-07967-f004:**
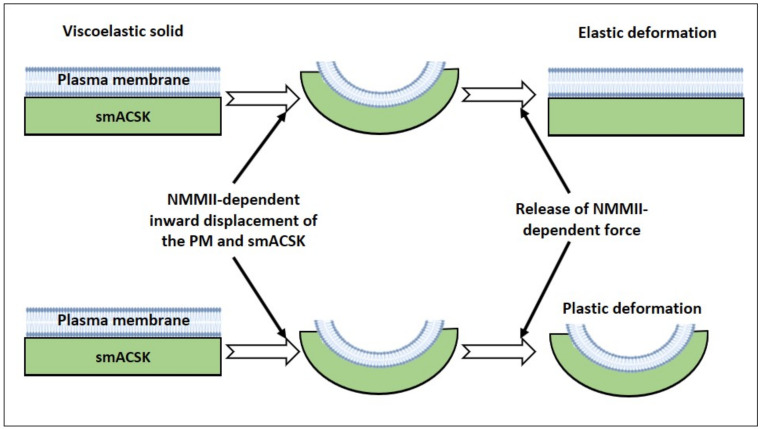
Illustration of behavior of a viscoelastic solid composed of the plasma membrane (blue rectangle) and the underlining submembrane cytoskeleton (green rectangle). Upper panel: in case of elastic deformation of the PM-smACSK under NMMII-generated pulling force the PM-smACSK completely restores its shape after the relaxation of NMMII. Lower panel: in case of the plastic deformation of the PM-smACSK, the PM-smACSK retains its deformed shape even after relaxation of NMMII.

**Figure 5 ijms-22-07967-f005:**
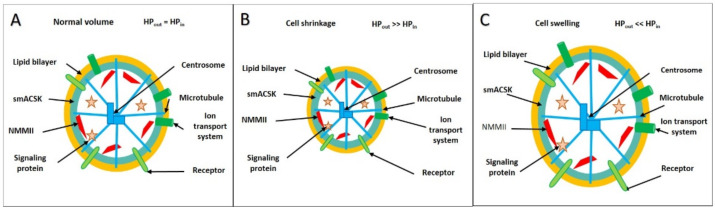
(**A**–**C**). Key players of cell volume sensing and regulation are NMMII (red dashes), the PM (yellow circle) together with the smACSK (blue-green circle) form the PM-smACSK complex, signaling proteins (red asters), cell surface receptors (green ovals) and ion transport systems (dark-green figures). The centrosome-based microtubular CSK (blue figures) forms the bulk CSK. Under normal cell volume, there is an equilibrium of hydrostatic pressure in the cell (HP_in_) and in the surrounding milieu (HP_out_) so that the shape of any local patch of the PM-smACSK can be assumed as planar (**A**). Under cell shrinkage, the PM-smACSK is being pulled into the cell (**B**). Under cell swelling, the PM-smACSK is being pushed into the extracellular space (**C**).

**Figure 6 ijms-22-07967-f006:**
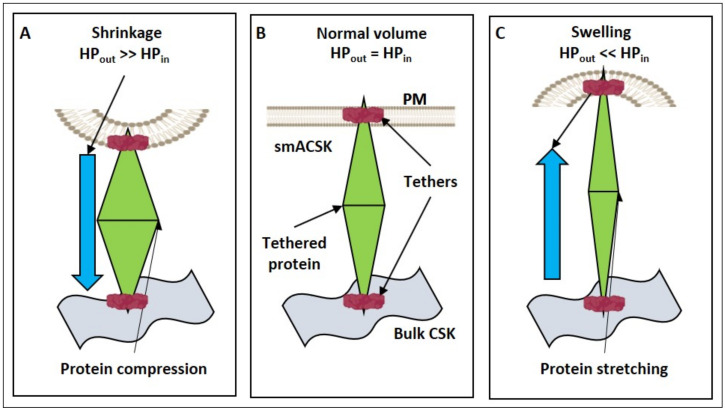
A transmembrane protein or membrane-associated protein (green diamond) is fixed by tethers (red ovals) at the PM-smACSK (orange and blue-green figures) and at the bulk CSK (grey wavy figure). (**A**): A mechanosensitive protein undergoes compression during cell shrinkage. (**B**): A mechanosensitive protein in normal cell volume. (**C**): A mechanosensitive protein undergoes stretching during cell swelling.

**Figure 7 ijms-22-07967-f007:**
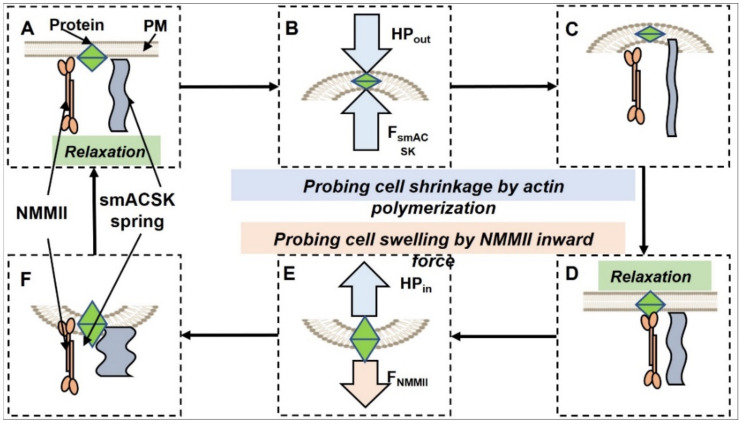
Scheme illustrating “two motors—one spring” model for probing of forces that drive displacements of the PM-smACSK out from or into the cell. (**A**–**C**): a force of F-actin protrusion drives the PM-smACSK out of the cell in order to probe the HP_out_ that in case of cell shrinkage drives the PM-smACSK into the cell. (**B**): A mechanosensitive signaling protein (green diamond) experiences compression that either activates or inhibits it. (**C**,**D**): After depolymerization of the protrusive actin filaments, the PM-smACSK restores its planar shape. (**D**–**F**): NMMII pulls the PM-smACSK into the cell and thus probes the increase in HP_in_ in case of cell swelling. (**E**): A mechanosensitive protein experiences distention that either activates or inhibits it. (**F**–**A**): NMMII is relaxed and the PM-smACSK spring restores its planar shape.

**Figure 8 ijms-22-07967-f008:**
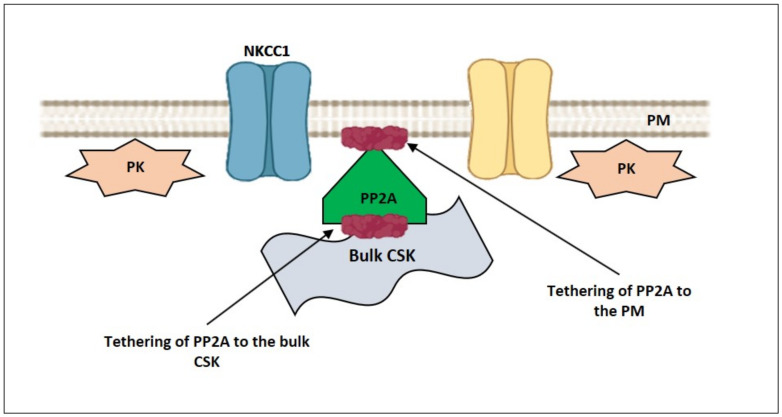
At normal cell volume NKCC1 and KCC are inactive, and their phosphorylation by proteins kinases and dephosphorylation by PP2A are balanced.

**Figure 9 ijms-22-07967-f009:**
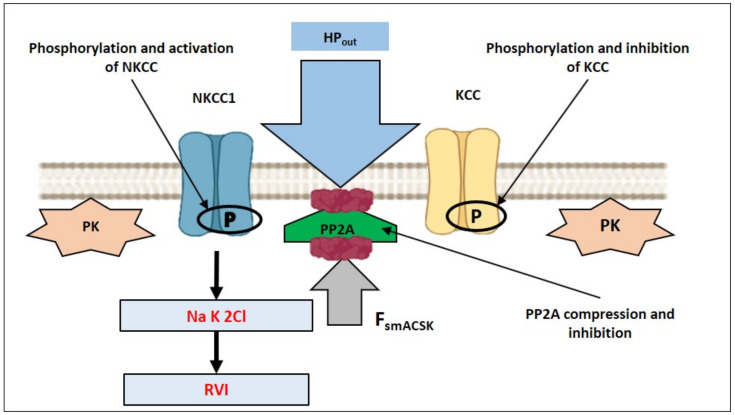
Illustration of hypothetical inactivation of mechanosensitive PP2A during cell shrinkage. PP2A can be compressed by both HP_out_ and F_smACSK_ and thus inactivated. Inactivation of PP2A would increase phosphorylation of NKCC1 by PKs that would activate NKCC1 and lead to RVI. In addition, phosphorylation of KCC by PKs would inactivate it.

**Figure 10 ijms-22-07967-f010:**
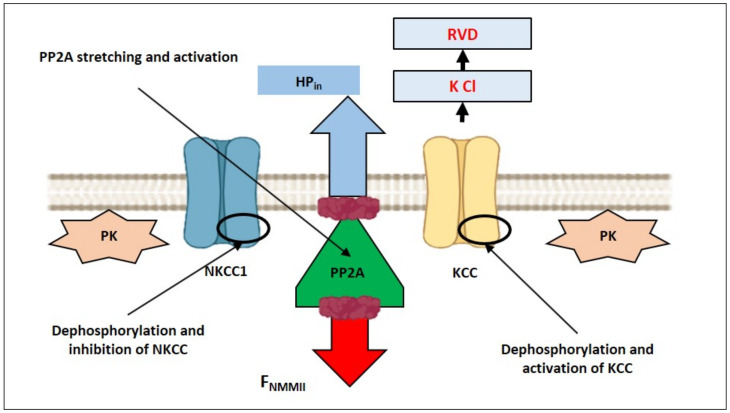
Illustration of hypothetical activation of mechanosensitive PP2A during cell swelling. During cell swelling PP2A can be stretched by HP_in_ and F_NMMII_ and thus activated. KCC would be activated by dephosphorylation by PP2A and would lead to RVD. In addition, dephosphorylation by PP2A would inhibit NKCC1.

## Data Availability

Not applicable.

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
