# Peer review of "Two Motors and One Spring: Hypothetic Roles of Non-Muscle Myosin II and Submembrane Actin-Based Cytoskeleton in Cell Volume Sensing"

_ijms, 2021, doi:10.3390/ijms22157967_

Round 1
Reviewer 1 Report
Manuscript IJMS-1298941: Two motors and one spring: Hypothetic roles of non-muscle myosin II and submembrane actin-based cytoskeleton in cell volume sensing”.
This paper describes a hypothetical apparatus for cell volume sensing. The authors describe the complex interplay of non-muscle myosin II and the actin-based cytoskeleton in an ATP-consuming mechanism.
Comments on the manuscript:
Section 2. Figure 1 in-box text should be black colored in order to enhance the visualization for the readers.
There should be a title, (which finds itself in legend for Figure 4) for subsection 2.3 (lines 251-252).
Section 3. Please check for consistency the sentence “These three elements can act together in measuring the strength of the hydrostatic pressure that does not displace the PM—smACSK under normal cell volume (Figure 5) but drives inward displacements of the PM—smACSK during cell shrinkage (Figure 6) and outward displacements during swelling (Figure 7).”
(lines 325-328) Should there be referring to Figure 5(B), 5(A) and 5(C) instead of referring to Figures 6 and 7?
Figure 7.F: the arrow from “smACSK spring” points to empty space and the arrow from 7.F to 7.A should shifted to the left.
Line 426, there should be HPin, not HPin.
Section 4. More attention should be paid to the effects of ionic strength on the submembrane spectrin-actin network, lines 462-464.
Section 6. The authors should mention the importance of cell volume sensing for clinical research, e.g., for brain and cancer research.
Author Response
We appreciate the reviewer's feedback and we would like to thank all the reviewers for the time spent on reviewing our manuscript.
We have carefully worked through all the reviewer’s comments and we have revised the manuscript accordingly. The specific changes made are listed below:
Reviewer 1:
Section 2. Figure 1 in-box text should be black colored in order to enhance the visualization for the readers.
Figure 1 has been modified accordingly.
There should be a title, (which finds itself in legend for Figure 4) for subsection 2.3 (lines 251-252).
We added a title for section 2.3 “Cell fates requiring volume change” accordingly
Section 3. Please check for consistency the sentence “These three elements can act together in measuring the strength of the hydrostatic pressure that does not displace the PM—smACSK under normal cell volume (Figure 5) but drives inward displacements of the PM—smACSK during cell shrinkage (Figure 6) and outward displacements during swelling (Figure 7).”
(lines 325-328) Should there be referring to Figure 5(B), 5(A) and 5(C) instead of referring to Figures 6 and 7?
We apologize for our mistake and have corrected it (lines 326-328)
Figure 7.F: the arrow from “smACSK spring” points to empty space and the arrow from 7.F to 7.A should shifted to the left.
Thanks for the suggestion, the figure has now been revised as suggested.
Line 426, there should be HPin, not HPin.
Corrected, now in line 428
Section 4. More attention should be paid to the effects of ionic strength on the submembrane spectrin-actin network, lines 462-464.
Section 4 has been improved by adding the following sentence and new references: “F-actin itself is negatively charged and can bind to positively charged lipids [96]. Numerous proteins that provide assembly of the actin CSK possess negatively and positively charged domains involved in protein-protein interactions. Palladin, an actin cross-linking protein, possesses two basic sequences on opposite sites of an immunoglobulin 3 domain which provide electrostatic interaction of paladin with two actin filaments [97].” Lines 463- 468
Section 6. The authors should mention the importance of cell volume sensing for clinical research, e.g., for brain and cancer research
As suggested by the reviewer we modified section 6 now 7, adding these statements and new references:
“The search for any particular volume-sensitive – or, as we assume here, active, energy-consuming cell volume sensing – molecular apparatus would be of importance for, e.g., anti-cancer treatment. Uncontrollable cell proliferation, resistance to apoptosis, migration – these features are among the hallmarks of cancer cell biology [149]. These cancer cell activities are critically dependent on volume regulatory systems, in particular on shifts in the volume set point. Ion channels are involved in the regulation of cell proliferation and apoptosis as well as in cell volume regulation [1,4,6–9]. NMMII and PM-smACSK could be a signaling hub linking execution of cell proliferation, migration and apoptosis to signaling pathways controlling activities of ion channels and transporters. NMMII isoforms appear to regulate the metastatic potential of cancer cells and, thus, to be novel therapeutic targets [150–152]. Our hypothesis on NMMII as a key player in cell volume sensing broadens the spectrum of effects of NMMII in cancer cells. (Patho)physiological roles of NHEs in organ systems — cardiovascular system, brain, kidneys, skeletal muscle, gastrointestinal tract, salivary glands, pancreas, eyes, immune system, reproductive system, bone, skin — were recently reviewed [9]. Our model suggests a mechanism for activation of NHE isoforms by cell shrinkage and swelling.” Lines 624-639
Reviewer 2 Report
The hypothesis formulated by Barvitenko et al. proposes a sub-membrane molecular apparatus, composed of NMMII and the PM-smCSK, as a sensor of the plasma membrane curvature changes, which subsequently triggers a series of intracellular pathways leading to RVI and RVD.
The manuscript is well structured, and easy to read. There are a total of 10 figures as support of the text, summarizing the main key points. The state of the art literature is up to date, well supporting the proposed model.
I have only few minor concerns:
(i) line 122, abbreviation of myosin light chain phosphatase seems to be omitted;
(ii) line 233, there should be consistent abbreviations and smACSK should stand here, not sm-CSK.
(iii) lines 325-328, There seems to be confusion in Figures’ mentioning in this sentence ’These three elements can act together in measuring the strength of the hydrostatic pressure that does not displace the PM—smACSK under normal cell volume (Figure 5) but drives inward displacements of the PM—smACSK during cell shrinkage (Figure 6) and outward displacements during swelling (Figure 7).’
Author Response
We appreciate the reviewer's feedback and we would like to thank all the reviewers for the time spent on reviewing our manuscript.
We have carefully worked through all the reviewer’s comments and we have revised the manuscript accordingly. The specific changes made are listed below:
Reviewer 2
- line 122, abbreviation of myosin light chain phosphatase seems to be omitted;
The abbreviation of myosin light chain phosphatase has now been added “MLCP” line 127
(ii) line 233, there should be consistent abbreviations and smACSK should stand here, not sm- CSK.
This has now been corrected.
- lines 325-328, There seems to be confusion in Figures’ mentioning in this sentence ’These three elements can act together in measuring the strength of the hydrostatic pressure that does not displace the PM—smACSK under normal cell volume (Figure 5) but drives inward displacements of the PM—smACSK during cell shrinkage (Figure 6) and outward displacements during swelling (Figure 7).’
We apologize for our mistake and have corrected it (lines 326-328).
Reviewer 3 Report
This hypothesis paper provides a potentially interesting, yet flawed, view on the relationship of the actin cortex with volume sensing in cells. In a way, the authors have very good intentions and a clear goal, which is to elaborate on the hypothesis that the actin cortex, actin protrusion and myosin II contractility work as a continuum elastic system that allows the cell to “sense” its own self-volume and control it.
A major problem is that the authors touch everything, everywhere, but in a cursory manner. Myosin II regulation? Check. Actin polymerization? Check. Membrane receptors? Check. However, none of these elements are discussed efficiently within the frame of the hypothesis.
Another major problem is that the paper is clearly written with a very artificial setup in mind, which is the induction of volume changes due to osmotic pressure. This is a setup that hardly comes to pass in physiological conditions. It wouldn't be an issue if everything else was exciting and well-discussed, which is not the case.
Another issue is that the paper has a clear biophysics vocation, yet no quantitative data is discussed. This is a field in which measurements (volume, forces, etc.) are of paramount importance.
Also, integrins are discussed in terms of their function as signal transducers, but in non-leukocytes, integrins can actually regulate cortex dynamics by providing anchorage points for adhesion, which alters many of the signals involved in volume sensing. In fact, adhesion-mediated cortex asymmetry is not discussed at all, and this is too widespread a tool not to consider it strongly when writing a general hypothesis paper such as this.
Another flaw is that the linkage of the actin to the plasma membrane is poorly discussed and is assumed to behave in a non-dynamic manner, i.e. remains constant. This is unrealistic, except in a few exceptional circumstances, e.g. blood cells; and even those are subject to many external forces in their physiological milieu that may affect mechanosensing.
Also, the paper funnels all the discussion on volume sensing towards the mechanical regulation of ion channel phosphorylation and mechanosensing via PP2A, which may be one mechanism involved, but it is hard to think it is the only one. No alternative mechanisms are included or discussed.
Finally, the actomyosin cortex is assumed to be homogeneous throughout the cell’s volume, which is highly unlikely (again, maybe in blood cells), but in reality, the cortex of most cells is inhomogeneous, asymmetric and displays spatial and temporal regulation in terms of PM linkage, growth and contraction.
I read this hypothesis paper with high expectations and really wanted it to be informative, but it comes out unfocused, too hypothetical and ignores too many aspects of the field to be considered a worthwhile contribution.
Author Response
Thank you for your comment and suggestion.
Reviewer 3
- This hypothesis paper provides a potentially interesting, yet flawed, view on the relationship of the actin cortex with volume sensing in cells. In a way, the authors have very good intentions and a clear goal, which is to elaborate on the hypothesis that the actin cortex, actin protrusion and myosin II contractility work as a continuum elastic system that allows the cell to “sense” its own self-volume and control it.
- A major problem is that the authors touch everything, everywhere, but in a cursory manner. Myosin II regulation? Check. Actin polymerization? Chec k. Membrane receptors? Check. However, none of these elements are discussed efficiently within the frame of the hypothesis.
We agree with the reviewer that many points have been dealt with in a cursory manner. However, the aim of this paper never was to provide a comprehensive review on all the elements that form part of our hypothesis, but to provide a novel hypothesis. It is outside the scope of an hypothesis paper to provide a detailed review of all mechanisms involved. These details are certainly provided in already published reviews, which we believe have been appropriately referenced in our paper.
- Another major problem is that the paper is clearly written with a very artificial setup in mind, which is the induction of volume changes due to osmotic pressure. This is a setup that hardly comes to pass in physiological conditions. It wouldn't be an issue if everything else was exciting and well-discussed, which is not the case.
We disagree with the characterisation “artificial”. Every hypothesis favors one possibility over another. As long as there is evidence provided that the hypothesis may be correct and as long as the hypothesis is testable, it may be wrong, but it is not artificial. Our hypothesis setup is based on the idea that cells are not passive sensors of any stress but rather active in probing their surroudings. We would like to stress that the scope of this manuscript is to provide the scientific community an hypothesis rather than a review.
In order to emphasize that this paper is an hypothesis, we have changed “Here” to “In this hypothesis” (line 25) and added the phrase “We present an hypothesis that comprises four key features…” (line 615).
- Another issue is that the paper has a clear biophysics vocation, yet no quantitative data is discussed. This is a field in which measurements (volume, forces, etc.) are of paramount importance.
Methods for direct measurements of forces that are acting on any single protein in live cell are yet under development. The aim of this manuscript is to provide a novel hypothesis that can eventually lead the application of already existing techniques on a different aspect.
- Also, integrins are discussed in terms of their function as signal transducers, but in non-leukocytes, integrins can actually regulate cortex dynamics by providing anchorage points for adhesion, which alters many of the signals involved in volume sensing. In fact, adhesion-mediated cortex asymmetry is not discussed at all, and this is too widespread a tool not to consider it strongly when writing a general hypothesis paper such as this.
- Another flaw is that the linkage of the actin to the plasma membrane is poorly discussed and is assumed to behave in a non-dynamic manner, i.e. remains constant. This is unrealistic, except in a few exceptional circumstances, e.g. blood cells; and even those are subject to many external forces in their physiological milieu that may affect mechanosensing.
By no means we assume the linkage of the cytoskeleton to the plasma membrane as a constant non-dynamic structure, on the contrary we emphasize trough all the manuscript, expecially in section 5.5, that actin cortex undergo remodeling depending on the cell program.
- Also, the paper funnels all the discussion on volume sensing towards the mechanical regulation of ion channel phosphorylation and mechanosensing via PP2A, which may be one mechanism involved, but it is hard to think it is the only one. No alternative mechanisms are included or discussed.
As there are multiple signaling proteins and other signlaing entities that participate in signaling regulating regulatory volume increase or decrease, we have chosen among them only one, PP2A, as an example. We have made it clear in the paper that PP2A is an example.
- Finally, the actomyosin cortex is assumed to be homogeneous throughout the cell’s volume, which is highly unlikely (again, maybe in blood cells), but in reality, the cortex of most cells is inhomogeneous, asymmetric and displays spatial and temporal regulation in terms of PM linkage, growth and contraction.
Thank you for your comment. We believe that high inhomogeneity of the actomyosin cortex will be of interest in later stage of validation of our hypothesis. For this paper the aim is to formulate the general hypothesis.
- I read this hypothesis paper with high expectations and really wanted it to be informative, but it comes out unfocused, too hypothetical and ignores too many aspects of the field to be considered a worthwhile contribution.
We are sorry that the reviewer does not like our hypothesis and obviously favors a different one. But then again, this may be the start of a fruitful scientific discusssion in the future. Again, we did not intent a comprehensive review, but a focused hypothesis.
Round 2
Reviewer 3 Report
The authors are convincing in one point. The use of the term artificial was not adequate. A more proper characterization is “far from physiological”.
The authors have elected not to change the manuscript significantly, hence the evaluation remains the same.